# Cool Temperature Enhances Growth, Ferulic Acid and Flavonoid Biosynthesis While Inhibiting Polysaccharide Biosynthesis in *Angelica sinensis*

**DOI:** 10.3390/molecules27010320

**Published:** 2022-01-05

**Authors:** Han Dong, Meiling Li, Ling Jin, Xiaorong Xie, Mengfei Li, Jianhe Wei

**Affiliations:** 1College of Pharmacy, Gansu University of Chinese Medicine, Lanzhou 730030, China; dongaa10245@163.com (H.D.); jinl@gszy.edu.cn (L.J.); 2State Key Laboratory of Aridland Crop Science, Gansu Agricultural University, Lanzhou 730070, China; mlli1996@163.com; 3Institute of Medicinal Plant Development, Chinese Academy of Medical Sciences and Peking Union Medical College, Beijing 100193, China; jhwei@implad.ac.cn

**Keywords:** *Angelica sinensis*, cool temperature, growth, ferulic acid biosynthesis, flavonoid biosynthesis, gene expression

## Abstract

*Angelica sinensis*, a perennial herb that produces ferulic acid and phthalides for the treatment of cardio-cerebrovascular diseases, prefers growing at an altitude of 1800–3000 m. Geographical models have predicted that high altitude, cool temperature and sunshade play determining roles in geo-authentic formation. Although the roles of altitude and light in yield and quality have been investigated, the role of temperature in regulating growth, metabolites biosynthesis and gene expression is still unclear. In this study, growth characteristics, metabolites contents and related genes expression were investigated by exposing *A. sinensis* to cooler (15 °C) and normal temperatures (22 °C). The results showed that plant biomass, the contents of ferulic acid and flavonoids and the expression levels of genes related to the biosynthesis of ferulic acid (*PAL1*, *4CLL4*, *4CLL9*, *C3H*, *HCT*, *CCO**AMT* and *CCR*) and flavonoids (*CHS* and *CHI*) were enhanced at 15 °C compared to 22 °C. The contents of ligustilide and volatile oils exhibited slight increases, while polysaccharide contents decreased in response to cooler temperature. Based on gene expression levels, ferulic acid biosynthesis probably depends on the CCOAMT pathway and not the COMT pathway. It can be concluded that cool temperature enhances plant growth, ferulic acid and flavonoid accumulation but inhibits polysaccharide biosynthesis in *A. sinensis*. These findings authenticate that cool temperature plays a determining role in the formation of geo-authentic and also provide a strong foundation for regulating metabolites production of *A. sinensis*.

## 1. Introduction

*Angelica sinensis* (Oliv.) Diels, commonly named as Dang gui, Dong quai and Tang kuei, is a perennial herb that grows at an altitude of 1800–3000 m with cool, moist and partial shade conditions [1,2]. The roots of *A. sinensis*, one of the most important herbal drugs in traditional Chinese medicine, are used for nourishing blood, regulating female menstrual disorders and relieving pains and relaxing bowels [1,3,4]. More recently, interest has focused on its potential cardio-cerebrovascular, hepatoprotective, antioxidant, antispasmodic and immunomodulatory effects [1]. Currently, over 140 constituents that have been identified from the roots mainly include the following: organic acids (e.g., ferulic acid, coniferyl ferulate and succinic acid), phthalides (e.g., ligustilide, butylidenephthalide and butylphthalide), polysaccharides (e.g., fucose, galactose and glucose) and flavonoids [4]. The main actives compounds in the roots are considered to be the following: organic acids, phthalides and polysaccharides [5].

*A. sinensis* is originally native to China with a population center in Gansu province as well as cultivated in western regions, including the following: Qinghai, Sichuan and Yunnan provinces [1,2,6]. Minxian county and surrounding counties (e.g., Tanchang, Zhangxian, Weiyuan, Zhuoni and Lintan) of Gansu province are known as the geo-authentic areas due to roots containing more bioactive compounds [2,7,8]. Analyses on geo-authentic *A. sinensis* grown in Gansu region have deduced that altitude, temperature, sunshine and rainfall are the most influential ecological factors for the accumulation of bioactive compounds (ferulic acid, ligustilide, chlorogenic acid, coniferyl ferulate, senkyunolide A, senkyunolide H, senkyunolide I, butenyl phthalide and levistilide A) [9,10]. Investigations into the role of altitude (*ca.* 2000–2900 m) have found that there is a positive correlation of altitude with the contents of ferulic acid, volatile oils (i.e., ligustilide and butenyl phthalide), polysaccharides, flavonoids and phenolics, while moderate altitude (*ca.* 2500–2600 m) is favorable for root biomass [11,12,13,14,15,16,17,18].

Generally, altitude affects climate by decreasing temperatures and increasing light intensity and rainfall. Previous studies have demonstrated that reducing light intensity with 50–75% of sunshade can increase root yield and ferulic acid accumulation, meanwhile indirectly decreasing temperature and increasing moisture of air and soil [19,20]; on the other hand, UV-B radiation can increase phthalide accumulation [21]. Although *A. sinensis* plants prefer moist conditions, the excessive soil water can induce fleshy roots to rot, and soil drought will result in a significant decrease in root yield as well as ferulic acid and volatile oils contents [1,22]. 

For the biosynthesis pathways of ferulic acid, flavonoids and phthalides, previous literature has reported that the biosynthesis of ferulic acid and flavonoids belongs to the phenylpropanoid pathway (Figure 1). Specifically, ferulic acid biosynthesis is synthesized via two sub-pathways, including the following: (1) caffeic acid 3-O-methyltransferase (COMT) pathway involved in enzymes such as cinnamate 4-hydroxylase (C4H), *p*-coumarate 3-hydroxylase (C3H) and COMT; and (2) caffeoyl-CoA 3-O-methyltransferase (CCOAMT) pathway involved in enzymes such as 4-coumarate-CoA ligase (4CL), hydroxycinnamoyl shikimate transferase (HCT) and CCOAMT. Flavonoid biosynthesis was catalyzed by enzymes such as chalcone synthase (CHS) and chalcone isomerase (CHI) [23,24]. On the other hand, the biosynthesis pathway of phthalides, especially in ligustilide, and the genes that participate in regulating the biosynthesis are still limited [9,25,26].

To date, plant growth, bioactive metabolites accumulation and related genes expression in *A. sinensis* in response to temperatures have not been examined. In this study, we probe the role of temperatures in growth, metabolites biosynthesis and genes expression related to ferulic acid, flavonoids, volatile oils and polysaccharides to identify links between cooler growing temperatures and the formation mechanism of geo-authentic medical materials of *A. sinensis*.

## 2. Results

### 2.1. Effect of Temperatures on Growth Characteristics

Fresh and dry weights (FW and DW) of whole plants were 1.19-fold and 1.51-fold greater at 15 °C than 22 °C, which largely relied on a 1.41-fold and 1.47-fold increase in FW and DW of roots (Figure 2A,B). Additionally, there was a 1.58-fold and 1.10-fold increase in stem and root diameters (Figure 2C,D), while no significant difference in shoot height and root length was observed between 15 °C and 22 °C (Figure 2E,F).

### 2.2. Effect of Temperatures on Contents of Ferulic Acid, Flavonoids, Ligustilide, Volatile Oils and Polysaccharides

Ferulic acid and flavonoids contents in roots were 1.90-fold and 1.42-fold greater at 15 °C than 22 °C (Figure 3A,B). Ligustilide and volatile oils contents exhibited a 1.01-fold and 1.15-fold increase at 15 °C, while there was no significant difference compared to 22 °C (Figure 3C,D). Polysaccharides content significantly decreased with a reduction of 0.86-fold at 15 °C compared to 22 °C (Figure 3E).

### 2.3. Effect of Temperatures on Gene Expression Related to Ferulic Acid and Flavonoid Biosynthesis

The mRNA expression levels of 10 genes related to ferulic acid biosynthesis (*PAL1*, *4CLL4*, *4CLL5*, *4CLL7*, *4CLL9*, *HCT*, *C3H*, *CCO**AMT*, *CCR1* and *COMT1*) and four genes related to flavonoid biosynthesis (*CHS*, *CHI*, *GT6* and *I3′H*) in roots at 15 and 22 °C were quantified. For ferulic acid biosynthesis, the relative expression levels (RELs) of seven genes, *PAL1*, *4CLL4*, *4CLL9*, *HCT*, *C3H*, *CCO**AMT* and *CCR1*, exhibited an upregulation of 8.05-fold, 3.65-fold, 2.74-fold, 2.50-fold, 11.48-fold, 10.82-fold and 3.60-fold, while the other three genes, *4CLL5*, *4CLL7* and *COMT1*, exhibited a downregulation of 0.65-fold, 0.35-fold and 0.18-fold, respectively, at 15 °C compared to 22 °C (Figure 4). For flavonoid biosynthesis, the RELs of the two genes *CHS* and *CHI* exhibited an upregulation of 4.16-fold and 3.65-fold, while the other two genes *GT6* (UDP-glucose flavonoid 3-O-glucosyltransferase 6) and *I3′H* (isoflavone 3′-hydroxylase) exhibited a downregulation of 0.95-fold and 0.40-fold, respectively, at 15 °C compared to 22 °C (Figure 5).

### 2.4. Effect of Temperatures on Gene Expression Related to Volatile Oils and Polysaccharide Biosynthesis

The mRNA expression levels of two genes related to volatile oils biosynthesis (trans-anol O-methyltransferase 1 (*AIMT1*) and acetyl-CoA-benzylalcohol acetyltransferase (*BEAT*)) and three genes related to polysaccharide biosynthesis (sucrose synthase isoform 1 (*SUS1*), pancreatic alpha-amylase (*Amy2*) and granule-bound starch synthase 1 (*WAXY*)) in roots at 15 and 22 °C were quantified. For volatile oil biosynthesis, the RELs of the two genes, *AIMT1* and *BEAT*, exhibited an upregulation of 1.11-fold and 1.10-fold, respectively (Figure 6). For polysaccharide biosynthesis, the RELs of the two genes *SUS1* and *WAXY* exhibited an upregulation of 8.91-fold and 5.16-fold, while gene *Amy2* exhibited a downregulation of 0.44-fold, respectively, at 15 °C compared to 22 °C.

## 3. Discussion

The formation of geo-authentic herbs not only lays on the human society and species but also natural environmental conditions [7,25]. Previous studies have found that the geo-authentic formation of *A. sinensis* depends on geographic environmental conditions (e.g., higher altitude, cooler temperature and less sunshine) [2,4,6,10]. In this study, we found that cooler temperatures enhanced root biomass, ferulic acid and flavonoids accumulation and related genes expression; and inhibited polysaccharide biosynthesis, while it did not significantly affect ligustilide accumulation. 

A significant increase in plant biomass was observed at cooler temperatures of 15 °C than 22 °C, which mainly resulted from the significant increase in root diameter (Figure 2). The increase in plant biomass at 15 °C authenticates that the *A. sinensis* species prefers cool environmental conditions, which is accordance with previous studies that higher-altitude improves the root yield and bioactive metabolites accumulation [11,12,14]. Several studies have found that cooler temperatures are conducive to plant growth and root biomass, such as increases in root diameter and biomass of *Sinopodophyllum hexandrum* seedings at 15 °C compared to 22 °C [15,26]; hairy roots biomass of *Panax ginseng* at 20 °C/13 °C compared to 25 °C and 35 °C/25 °C [27]; and whole plant biomass of *Hypericum* perforatum at 15 °C compared to 22 °C [28]. 

The accumulation of plant secondary metabolites is often affected by environmental factors, such as light, water and temperature [29]. Low temperatures are one of the most important factors regulating phenylpropanoid metabolism [30]. Both ferulic acid and flavonoid biosynthesis pathways belong to the phenylpropanoid metabolism and employ the same genes, *PAL*, *C4H* and *4CL* (Figure 1). In this study, a significant increase in ferulic acid and flavonoids contents was observed at 15 °C compared to 22 °C (Figure 3A,B), which largely relied on upregulation of genes related to ferulic acid (i.e., *PAL1*, *4CLL4*, *4CLL9*, *C3H*, *HCT*, *CCO**AMT* and *CCR*) and flavonoid biosynthesis (i.e., *CHS* and *CHI*) (Figure 1, Figure 4 and Figure 5). A significant downregulation of *COMT1* (Figure 4) indicates that ferulic acid biosynthesis depends on not COMT but the CCOAMT pathway (Figure 1).

In addition, with respect to the two genes *4CLL5* and *4CLL7* that were downregulated at 15 °C compared to 22 °C, the gene *4CLL5* was found to contribute to jasmonic acid biosynthesis [31], and gene *4CL7* encodes enzyme 4CL7 that had no catalytic activity toward hydroxycinnamic acid compounds [32], which indicate that two genes *4CLL5* and *4CLL7* do not participate in the ferulic acid biosynthesis. For the other two genes *GT6* and *I3′H* that were downregulated at 15 °C compared to 22 °C, gene *GT6* is involved in xenobiotic metabolism [33], and gene *I3*′*H* is involved in the biosynthesis of pterocarpan phytoalexins [34]; both *GT6* and *I3*′*H* are required for pathogen defense and insect-induced responses [35,36]. The downregulation of two genes *GT6* and *I3′H* further confirms that the *A. sinensis* species is an alpine plant that prefers a cool environment.

No significant increase in ligustilide and volatile oils contents was observed at 15 °C compared to 22 °C (Figure 3C,D), which is consistent with the slight upregulation of two genes *AIMT1* and *BEAT* involved in volatile oils biosynthesis (Figure 6). For the biological functions, gene *AIMT1* is involved in the conversion of anethole and isoeugenol to isomethyleugenol, which are the primary constituents of volatile oils [37], and gene *BEAT* is involved in the biosynthesis of benzyl acetate [38]. 

For the polysaccharide accumulation, a significant decrease was observed at 15 °C compared to 22 °C (Figure 3E), while the three genes *SUS1*, *Amy2* and *WAXY* were observed to be differentially regulated at 15 °C, with upregulation for *SUS1* and *WAXY* and, on the other hand, downregulation for *Amy2* (Figure 6). For biological functions, gene *SUS1* is involved in sucrose cleaving, which provides UDP-glucose and fructose for various metabolic pathways such as glycolysis [39]; gene *WAXY* is involved in starch and glycan biosynthesis [40]; and the gene *Amy2* is involved in starch hydrolase [41]. In this study, the upregulation of the gene *SUS1* may degrade polysaccharides to glucose and fructose, which provide energy to adapt to cooler temperatures. The upregulation of gene *WAXY* and downregulation of gene *Amy2* may promote starch accumulation, which results in greater plant biomass at 15 °C than 22 °C.

## 4. Materials and Methods

### 4.1. Plant Material

Seedlings of *Angelica sinensis* (cultivar Mingui 1) with root-tip diameter 0.4–0.5 cm (see Appendix A) were selected to plant in pots (13 cm × 9 cm) with soil (coconut coir: peat: fermented cow dung: pearlite = 3:3:2:2) and to germinate in a growth chamber with a constant temperature (18 °C) and a photoperiod cycle (16/8 h light/dark, 500 µmol·m^−2^·s^−1^). After 15 days, plantlets that contained two leaves (see Appendix A) were moved to a growth chamber set at a constant temperature of 15 or 22 °C. After 30 days growth (see Appendix A), plants were harvested for physiological measurement, metabolites determination and mRNA quantification. Herein, higher temperatures such as 30 °C is excluded from treatments because *A. sinensis* is an alpine plant that prefers a cool environment with average annual temperatures ranging from 4 to 9 °C [1,2,9].

### 4.2. Physiological Measurement

After temperature-treated plants were removed from pots and rinsed with tap water, shoot height (cm), stem diameter (mm), root length (cm), diameter (mm), fresh weight (FW, g) and dry weight (DW, g) of aerial parts and roots were measured.

### 4.3. Metabolites Determination

#### 4.3.1. Extracts Preparation

After air-dried roots, finely powdered aliquots (0.2 g) were soaked in ethanol (95% *v*/*v*, 20 mL) and agitated at 25 °C and 120 r/min for 72 h. The homogenate was centrifuged (TGL-20M, Changsha, China) at 4 °C and 5000 r/min for 10 min. The extracts were increased to 25 mL with ethanol (95% *v*/*v*) for determination of ferulic acid, ligustilide, flavonoids and polysaccharides.

#### 4.3.2. Determination of Ferulic Acid and Ligustilide Contents

Ferulic acid and ligustilide contents were determined according to the previous protocol [42]. Briefly, extracts (10 μL) were determined at 323 nm using an HPLC Symmetry^®^ C_18_ column (250 mm × 4.6 mm, 5 μm; column temperature 30 °C; Waters ACQUITY Arc, Milford, MA, USA). The solution of acetic acid (1.0% *v/v*, A)-acetonitrile (B) was the mobile phase with gradient elution: 38–90% B (0–8 min), 90–38% B (8–12 min) and 38% B (12–14 min) at a flow rate of 1.0 mL/min. Ferulic acid and ligustilide contents were evaluated on peak area comparison with a reference standard. Representative HPLC chromatograms of standard reference and samples at 15 and 22 °C were shown in Appendix A.

#### 4.3.3. Determination of Flavonoids Content

Flavonoids content was determined using the NaNO_2_-AlCl_3_-NaOH method [43,44]. Briefly, the extracts (2.5 mL) were added into ddH_2_O (2 mL) and NaNO_2_ (5% *w*/*v*, 0.3 mL); after oscillation, and AlCl_3_ (10% *w*/*v*, 0.3 mL) was added and reacted at 22 °C for 1 min; then, NaOH (1.0 mol/L, 2 mL) was added to stop the reaction. An absorbance reader was taken at 510 nm using a spectrometer (V1800, Shanghai, China). Flavonoid content was expressed as milligram of catechin.

#### 4.3.4. Determination of Polysaccharides Content

Polysaccharides content was determined using the sulfuric acid-phenol protocol method [45,46]. Briefly, the extracts (150 µL) were added into a phenol reagent (9% *v*/*v*, 1 mL); after oscillation, sulfuric acid (3 mL) was added and reacted at 22 °C for 30 min. An absorbance reader was taken at 485 nm by using a spectrometer (V1800, Shanghai, China). Polysaccharides content was expressed as milligram of sucrose.

#### 4.3.5. Volatile Oils Determination

The extract of volatile oils was conducted using a steam distillation method [47]. Briefly, air-dried roots powder (5.0 g) was soaked in dH_2_O (30 mL) and extracted in a steam distillation apparatus for 8 h; after NaCl (1.0 g) added into the extracts and left standing for 10 min, ethyl acetate (30 mL) was added; following exhaustive extraction (×3), the upper portion was pooled, filtered and dried in vacuo at 35 °C to evaporate ethyl acetate. The rate of volatile oils was expressed as extract volume (mL) of roots weight (g).

### 4.4. Quantification of mRNA

Total RNA was extracted from roots using a Plant RNA Kit (R6827, Omega Bio-Tek, Inc., Norcross, GA, USA). The quality of the total RNA was examined using 1.0% agarose gel electrophoresis. Based on RNA sequencing and analysis of bolting and flowering of *A. sinensis* in our previously published articles [48,49], 19 candidate genes involved in bioactive metabolite biosynthesis were dig out. Primer sequences for 19 candidate genes (Table 1) were designed using an NCBI Primer-BLAST tool. cDNA was synthesized using a FastKing RT kit (KR116, Tiangen, China), and qRT-PCR was performed by ABI QuantStudio 5 system (USA) with a SuperReal PreMix Plus (SYBR Green) (FP205, Tiangen, China). *Actin* was used as an internal reference. The relative expression level (REL) was evaluated based on a 2^−^^△△Ct^ method [50].

### 4.5. Statistical Analysis

All measurements were performed using three biological replicates. A *t*-test in SPSS 22.0 was performed for independent treatments with *p* < 0.05 as the basis for statistical differences.

## 5. Conclusions

From the above observations, cooler temperatures significantly enhance biomass accumulation, ferulic acid and flavonoid biosynthesis in *A. sinensis* as well as their related genes expression while inhibiting polysaccharide accumulation. These findings will provide a strong foundation for regulating plant growth and bioactive metabolites production of *A. sinensis*. The roles of sunshade and rainfall in the geo-authentic formation will be conducted in the upcoming studies.

## Figures and Tables

**Figure 1 molecules-27-00320-f001:**
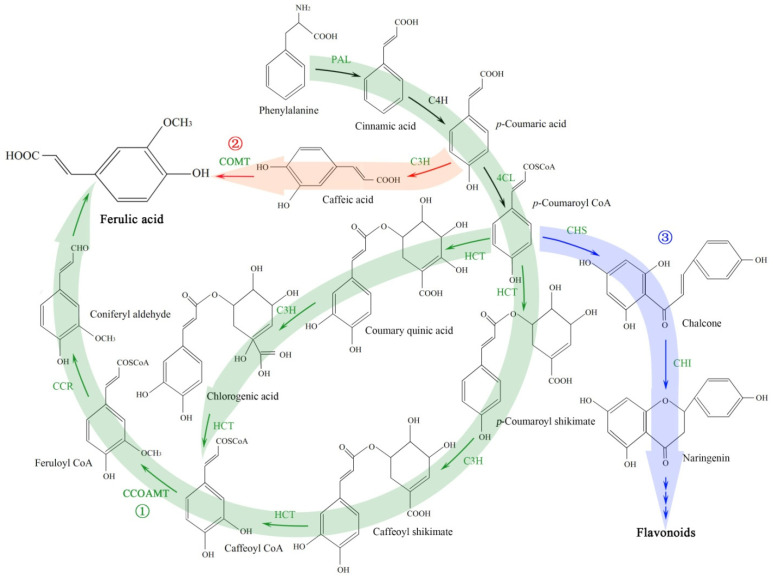
Schematic representation of biosynthetic pathways leading from shikimic acid pathway to phenylpropanoid pathway. Solid arrow indicates known steps, whereas multiple arrows indicate multiple reaction steps. Enzyme abbreviations are as follows: EMB3004, Bifunctional 3-dehydroquinate dehydratase/shikimate dehydrogenase, chloroplastic; CM, chorismate mutase; PAL, phenylalanine ammonia lyase; C4H, cinnamate 4-hydroxylase; 4CL, 4-coumarate-CoA ligase; HCT, hydroxycinnamoyl shikimate transferase; C3H, *p*-coumarate 3-hydroxylase; CCOAMT, caffeoyl-CoA 3-O-methyltransferase; CCR, cinnamoyl CoA oxidoreductases; COMT, caffeic acid 3-O-methyltransferase; CHS, chalcone synthase; CHI, chalcone isomerase. ① Showing the ferulic acid biosynthesis via CCOAMT sub-pathway; ② Showing the ferulic acid biosynthesis via COMT sub-pathway; ③ Showing the flavonoid biosynthetic sub-pathway.

**Figure 2 molecules-27-00320-f002:**
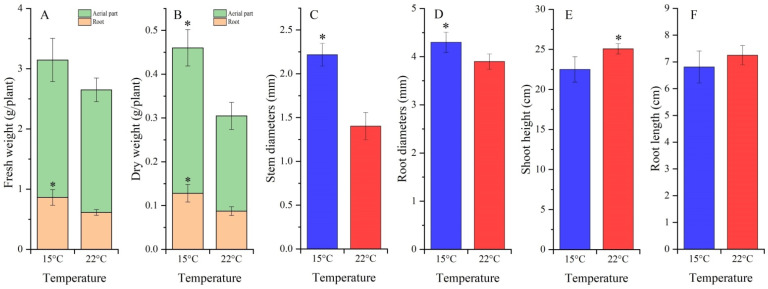
Growth characteristics in *A. sinensis* treated with cool temperature (mean ± SD, n = 20). Images (**A**,**B**) represent the FW and DW of the entire plant, (**C**,**D**) represent the stem and root diameters and (**E**,**F**) represent the shoot height and root length. “*” represents a significant difference (*p* < 0.05) at different stages. The same below.

**Figure 3 molecules-27-00320-f003:**
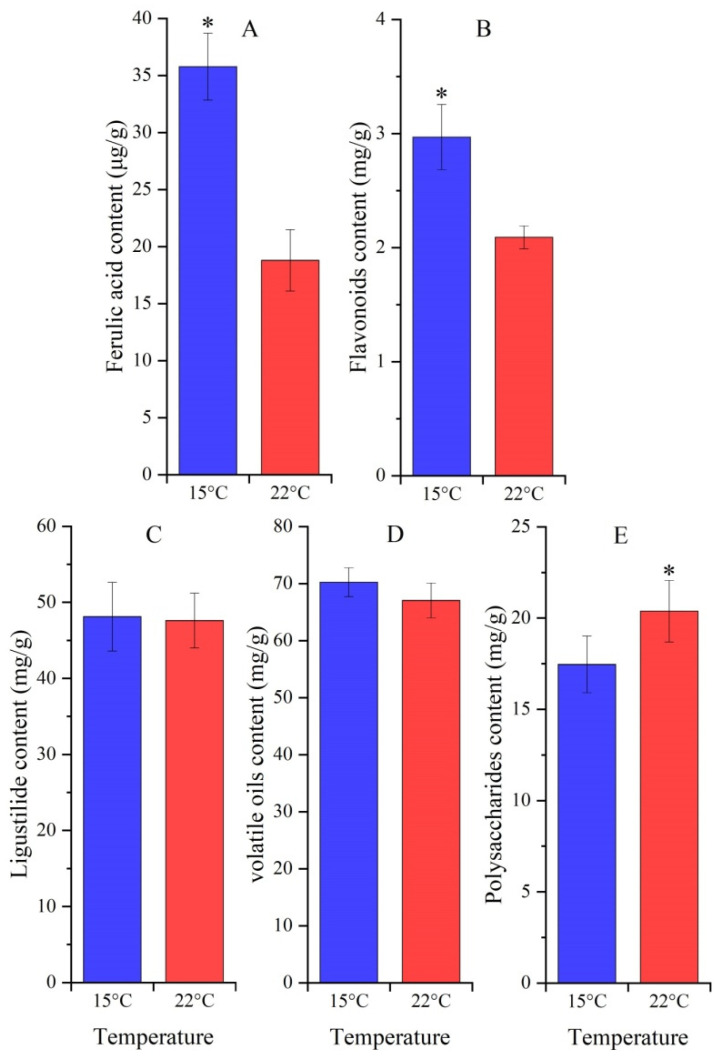
Metabolites contents in *A. sinensis* treated with cool temperature (mean ± SD, n = 20). Images (**A**,**B**) represent the ferulic acid and flavonoids contents, (**C**,**D**) represent the ligustilide and volatile oils contents and (**E**) represents the polysaccharides content, respectively. “*” represents a significant difference (*p* < 0.05) at different stages.

**Figure 4 molecules-27-00320-f004:**
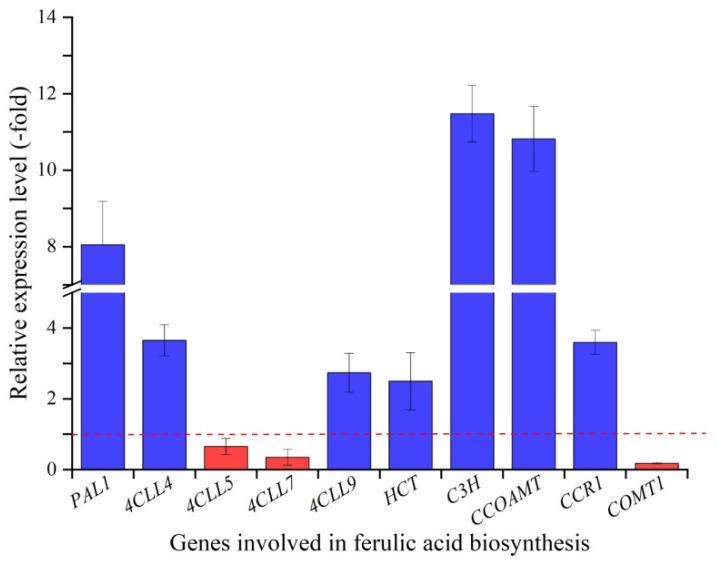
Expression levels of ten genes related to ferulic acid biosynthesis in *A. sinensis*. Histograms show the relative expression level in response to 15 °C compared to 22 °C. The same below. Abbreviations: PAL1, phenylalanine ammonia lyase 1; 4CLLs, 4-coumarate-CoA ligase like proteins; HCT, hydroxycinnamoyl shikimate transferase; C3H, *p*-coumarate 3-hydroxylase; CCOAMT, caffeoyl-CoA 3-O-methyltransferase; CCR1, cinnamoyl CoA oxidoreductase 1; COMT1, caffeic acid 3-O-methyltransferase 1.

**Figure 5 molecules-27-00320-f005:**
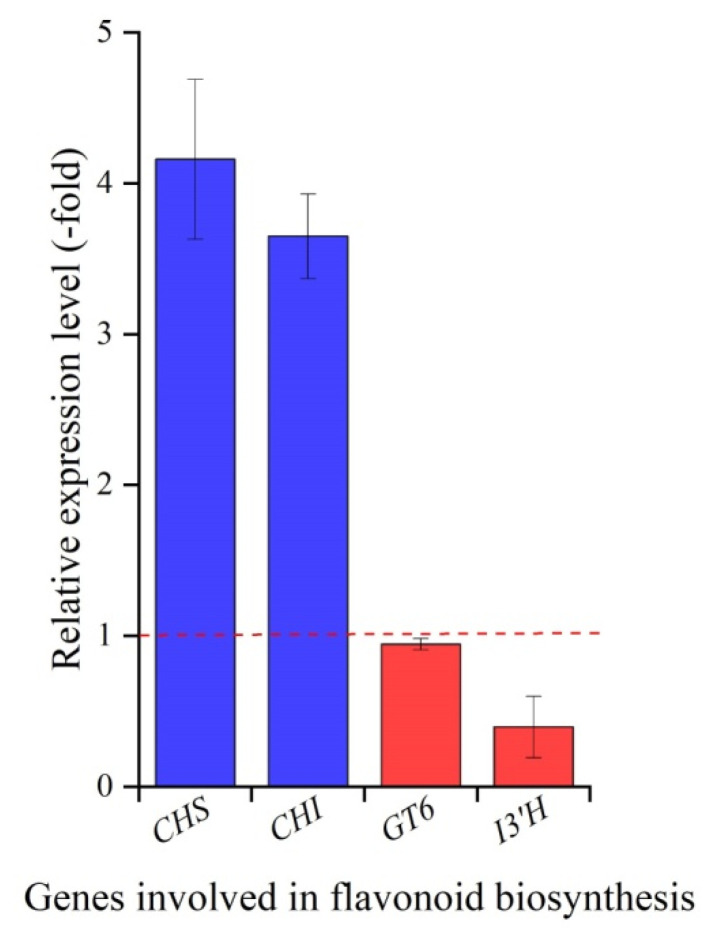
Expression levels of four genes related to flavonoid biosynthesis in *A. sinensis*. Abbreviations: CHS, chalcone synthase; CHI, chalcone isomerase; GT6, UDP-glucose flavonoid 3-O-glucosyltransferase 6; I3′H, isoflavone 3′-hydroxylase.

**Figure 6 molecules-27-00320-f006:**
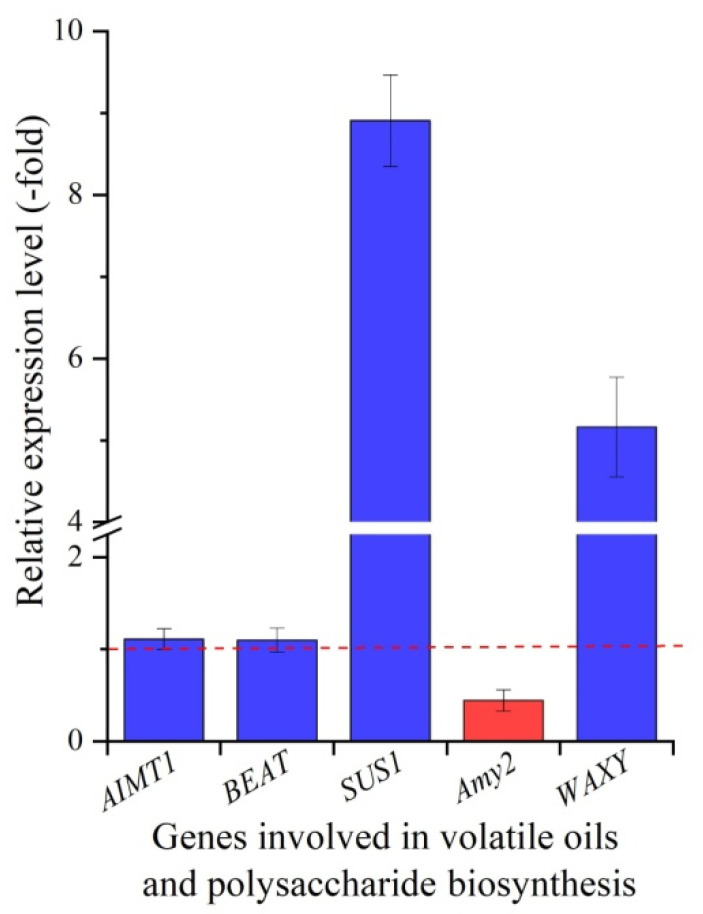
Expression levels of five genes related to volatile oils and polysaccharide biosynthesis in *A. sinensis*. Abbreviations: AIMT1, trans-anol O-methyltransferase 1; BEAT, acetyl-CoA-benzylalcohol acetyltransfe-rase; SUS1, sucrose synthase isoform 1; Amy2, pancreatic alpha-amylase; WAXY, granule-bound starch synthase 1.

**Table 1 molecules-27-00320-t001:** Primer sequences used in qRT-PCR analysis.

Genes	Accession No.	Sequences (5′ to 3′)	Amplicon Size (bp)
*ACT*	[26]	Forward: TGGTATTGTGCTGGATTCTGGT	109
		Reverse: TGAGATCACCACCAGCAAGG	
**Ferulic acid Biosynthesis**			
*PAL1*	XM_017399483.1	Forward: GGACTTGACAGTAGGGCAG	146
		Reverse: CCCCGTAACTATCCGTTCCTT	
*4CLL4*	XM_017376722.1	Forward: AAGCAGTGTTTCAGAGGCAG	105
		Reverse: GCTGAGCGCGGTATTGAGTT	
*4CLL5*	XM_017388768.1	Forward: CGGGACGAGTAAAGGAGTGG	171
		Reverse: AGCGTTGCTACAAACCAAGC	
*4CLL7*	KJ531407.1	Forward: TGCTCCGTTGGGTAGAGAGT	164
		Reverse: CTCCAGGCACAAGCATTCCT	
*4CLL9*	XM_017397573.1	Forward: GGTGGGGAAGCTAACAGGTC	183
		Reverse: TCGCCAGTTCTTAACCAGCC	
*HCT*	XM_017397289.1	Forward: CCGGTGACATATCTGCGTGT	171
		Reverse: GCGGAATGGCAATGGAAAGG	
*C3H*	[26]	Forward: CAATCCAAGTTGACGACGAA	119
		Reverse: CGAAGGCGAAACATAGGC	
*CCOAMT*	AY620245.1	Forward: TCGGCTACGACAACACCCTA	157
		Reverse: TCGCCAACAGGAAGCATACA	
*CCR1*	XM_017403617.1	Forward: CCATTCATGGATGCGTTGGT	135
		Reverse: CCACACGTCTCACATTGGCT	
*COMT1*	XM_010673030.2	Forward: TGGCGGAAAGGTAGTCGTTG	130
		Reverse: TTCAGTCCTCTCACTTCCGC	
**Flavonoid Biosynthesis**			
*CHS*	KP726914.1	Forward: GCAAAGACGCTGCATCCAAA	126
		Reverse: GGAGCTTGGTGAGCTGGTAG	
*CHI*	XM_017365109.1	Forward: GTGTTTCCCCAGCTGCAAAG	102
		Reverse: TTCCGACTTCTGCTTTCCCA	
*I3′H*	XM_017363227.1	Forward: GGCCACCTTCACCTCATCAA	173
		Reverse: GGGCGGTCAGCTAAAACAAC	
*GT6*	XM_017383880.1	Forward: TTCGGTGCCCATCACAAGAA	166
		Reverse: AATCCTCCGACAGATGCGTG	
**Volatile oils Biosynthesis**			
*AIMT1*	B8RCD3.1	Forward: CGCTAGTCTTTTGAGCGAAGC	119
		Reverse: CATGGGCACCTCCTACATCC	
*BEAT*	O64988.1	Forward: GATCAAGCCAGCAGTGATGC	147
		Reverse: ACTTCAACACGTGTAGGCCG	
**Polysaccharide Biosynthesis**			
*SUS1*	XM_017363708.1	Forward: ATGAAGTCCACACAGGAAGCC	124
		Reverse: CGACGACAAGGTGATGAGTG	
*Amy2*	V00718.1	Forward: TCTTCTGAGCCCTGGAGTGT	117
		Reverse: TCCAGGGAAGCCTCATGGAT	
*WAXY*	AJ006293.1	Forward: GCACTCATCCTCCATTCAGAG	167
		Reverse: TCCGTTACTGATCCACCAGC	

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
