# Peer review of "Cool Temperature Enhances Growth, Ferulic Acid and Flavonoid Biosynthesis While Inhibiting Polysaccharide Biosynthesis in Angelica sinensis"

_molecules, 2022, doi:10.3390/molecules27010320_

Round 1

Reviewer 1 Report

Review of the article Han Dong et al. «Cool Temperature Enhances Growth, Ferulic Acid and Flavonoid Biosynthesis while Inhibits Polysaccharide Biosynthesis  in Angelica sinensis»

The article is devoted to effect of low temperature on some physiological and biochemical parameters in Angelica sinensis. The topic of scientific work is relevant. The authors use modern research methods in their work. The article contains new data.

Remarks: 

In Introduction:

  • Line 90 “.. and the formation mechanism of geo-authentic herb  sinensis.”

Apparently,  “.. and the formation mechanism of geo-authentic medical materials of A. sinensis?”

In Results:

  • Judging by Fig. 2 F, the root length in plants at temperatures of 15° C and 22° C practically did not differ. Therefore, it would be correct to say that the temperature of 15° C does not affect root growth.
  • 2 E: Not plant height, but shoot height.

In Discussion:

  • How can you explain the increase of root diameter in plants at a temperature of 15° C?
  • Lines 187-189 “ The downregulation of two genes GT6 and I3’H indirectly shows that cooler temperature is not a stress condition but a suitable environment for sinensis.  However, it was initially  known that this plants species is an alpine that prefers to a cool environment.

In Materials and Methods:

  • Line 271. It is necessary to clarify what is meant by "three biological replicates". Are these three plants in each variant? Or are they three replicates of each variant of the experiment?
  • What was the analytical replication for the determination of biochemical parameters?

In Conclusions:

  • Line 275 “… cooler temperature significantly enhances growth,.. “ This is not entirely correct. The root length at 15 ° C and 22 ° C was almost the same, and the shoot height was higher at 22 ° C. More correct “…enhances biomass accumulation,..”

In general, the article may be published after minor corrections.

27.12.2021

Author Response

Reviewer: 1

The article is devoted to effect of low temperature on some physiological and biochemical parameters in Angelica sinensis. The topic of scientific work is relevant. The authors use modern research methods in their work. The article contains new data.

In Introduction:

Line 90 “and the formation mechanism of geo-authentic herb A. sinensis.” Apparently, “and the formation mechanism of geo-authentic medical materials of A. sinensis?”

According to your comments, the description has been revised to “the formation mechanism of geo-authentic medical materials of A. sinensis”. (Page 3, line 94)

In Results: Judging by Fig. 2 F, the root length in plants at temperatures of 15°C and 22°C practically did not differ. Therefore, it would be correct to say that the temperature of 15°C does not affect root growth.

In Results: Fig. 2 E: Not plant height, but shoot height.

According to your comments, the words “plant height” has been revised to “shoot height”, meanwhile, the words in the Fig. 2E as well as throughout the text has also revised. (Page 3, lines 100-102; Page 8, line 227)

The description on plant height and root length has been corrected to “no significant difference in shoot height and root length was observed between 15°C and 22°C”. (Page 3, line 101)

In Discussion: How can you explain the increase of root diameter in plants at a temperature of 15°C?

The description “the increase of root diameter in plants at a temperature of 15°C” in the Discussion section is coming from the investigation results in the Results section Fig. 2D, and the description has been revised to: “which mainly resulted from the significant increase in root diameter”. (Page 7, line 166

Lines 187-189 “The downregulation of two genes GT6 and I3’H indirectly shows that cooler temperature is not a stress condition but a suitable environment for sinensis. However, it was initially known that this plants species is an alpine that prefers to a cool environment.

Indeed, the Angelica sinensis species is an alpine plant that prefers to a cool environment.

Thanks for your comments, the sentence has been revised to: “The downregulation of two genes GT6 and I3’H further confirms that the A. sinensis species is an alpine plant that prefers to a cool environment”. (Page 8, lines 194-195)

In Materials and Methods:

Line 271. It is necessary to clarify what is meant by "three biological replicates". Are these three plants in each variant? Or are they three replicates of each variant of the experiment? What was the analytical replication for the determination of biochemical parameters?

According to your comments, the description “three biological replicates” has been clarified by adding the “(mean ± SD, n =20)” in the title Fig. 2 and Fig. 3. (Page 4, line 104 and Page 5, line 115)

In Conclusions:

Line 275 “… cooler temperature significantly enhances growth,.. “ This is not entirely correct. The root length at 15°C and 22°C was almost the same, and the shoot height was higher at 22°C. More correct “…enhances biomass accumulation...”.

Thanks for your suggestion, the description “---enhances biomass accumulation---” has replaced of “--- enhances growth---”. (Page 10, lines 282-284)

Reviewer 2 Report

The Manuscript is well presented and can be accepted after moderate revisions.

Kindly address all the points raised in the attached PDF file.

Minor improvement in the English language is required.

The quality of the figures may be enhanced if possible.

Author Response

Reviewer: 2

In Abstract:

Lines 14 to 18 “Although geographical models have predicted ... and gene expression is still unclear” too long sentence. Better rephrase.

According to your comments, the long sentence has been rephrased to: “Geographical models have predicted that high altitude, cool temperature and sunshade play determining roles in the geo-authentic formation. Although the roles of altitude and light in yield and quality have been investigated, the role of temperature in regulating growth, metabolites biosynthesis and gene expression is still unclear”. (Page 1, lines 14-18)

Lines 25 to 27 “Additionally, the ferulic acid biosynthesis ... the genes expression levels.” need to rephrase.

According to your comments, “Additionally, the ferulic acid biosynthesis ... the genes expression levels.” have been rephrased: “Based on the genes expression levels, the ferulic acid biosynthesis probably depends on the CCOAMT pathway not the COMT pathway”. (Page 1, lines 24-26)

In Introduction:

Must mention the common name of Angelica sinensis as well.

According to your comments, the common name of Angelica sinensis has been added: “commonly named as Dang gui, Dong quai, and Tang kuei”. (Page 1, lines 35-36)

Line 43 “The main bioactive compounds are pharmacologically considered to be: organic acids, phthalides and polysaccharides” need to be rephrased.

According to your comments, the sentence has been rephrased to: “The main actives compounds in the roots are considered to be: organic acids, phthalides and polysaccharides”. (Page 2, lines 45-46)

In Discussion:

Line 159 “at cooler temperature 15°C than 22°C” should add word “of ”.

Thanks for your suggestion, the word “of” has been added in the sentence. (Page 7, line 165)

Lines 172 to 176, “In this study----- flavonoid biosynthesis” need to be rephrased.

According to your comments, the sentence has been rephrased to: “In this study, a significant increase of ferulic acid and flavonoids contents was observed at 15°C compared to 22°C (Fig. 3A and 3B), which largely relied on upregulation of genes related to ferulic acid (i.e. PAL1, 4CLL4, 4CLL9, C3H, HCT, CCOAMT and CCR) and flavonoid biosynthesis (i.e. CHS and CHI) (Fig. 1, Fig. 4 and Fig. 5)”. (Page 7, lines 179-183)

Line 179 , the world “for” should be deleted.

The word “for” has been deleted. (Page 8, line 186)

In Materials and Methods:

Line 211 “germinate at --- 16/8 h light/dark cycle” need to be rephrased.

According to your comments, the sentence has been rephrased to: “---germinate at a growth chamber with a constant temperature (18°C) and a photoperiod cycle (16/8 h light/dark, 500 µmol•m-2•s-1)”. (Page 8, lines 217-218)

Line 221 “the shoot height --- roots were measured”, do mention the units of each of the parameter here in parenthesis.

According to your comments, the units of each of the parameter has been mentioned: “---shoot height (cm), stem diameter (mm), root length (cm) and diameter (mm), fresh weight (FW, g) and dry weight (DW, g) of aerial parts and roots were measured”. (Page 8, lines 227-228)

Reviewer 3 Report

The manuscript in titled (Cool Temperature Enhances Growth, Ferulic Acid and Flavonoid Biosynthesis while Inhibits Polysaccharide Biosynthesis 3 in Angelica sinensis) contains a very good results but the discussion and conclusion parts are very weak and needs to rewrite. So, I will be choose Major revision to can authors improve their manuscript 

There are some point must be improved:

1- Figure 1 not clear

2- There are some  abbreviations need to identify see the attachment file

3- some minor language issues needs to improve

4- The discussion part very weak and needs to rewritten. It is only  a return to the introduction and the results without explanations or reasons

5- where is the centrifuge company and country made in line 226 page 7

6- where is the spectrometer company and country made in line 244 page 8

7- Very weak conclusion, It must be one sentence to the most important results, sentence to results applications and whom will benefits from these results. A future outlook in the field of research, upcoming studies, and the beneficiaries thereof

8- The conclusion part never has any references

Author Response

Reviewer: 3

In Introduction:

Figure 1. is not clear figure and please identify the abbreviations of “COMT” and “CCOMT”.

Thanks for your carefully reviewing, the abbreviations of “CCOMT” has been written by mistake, now it has been corrected to “CCOAMT” . In order to show the Fig. 1 clearly, the Fig. 1 has been enlarged. (Page 3, lines 86 and 87)

In Results:

The world “ upregulation” should be preceded by “an ”, not “a”.

Thanks for your kind reminding, the word has been corrected. (Page 5, lines 123 and 127; Page 7, lines 149 and 151)

In all figures identify all the abbreviations in the footnotes under figures.

According to your comments, all the abbreviations in the figures have been added in the footnotes under figures. (Page 6, lines 133-136 and 139-141; Page 7, lines 154-156)

In Discussion:

Lines 149 to 153 “Extensive literatures---quality of A. sinensis”, its repeated sentence exchange it with novel one.

Lines 154 to 158 “Specifically---ligustilide accumulation”, repeated in introduction also.

According to your comments, the repeated sentences have been deleted or revised to: “Previous studies have found that the geo-authentic formation of A. sinensis depends on the geographic environmental conditions (e.g. higher altitude, cooler temperature and less sunshine) [2, 4, 6, 10]”. (Page 7, lines 159-161)

Lines 159 to 167 “Significant increase---the cool environmental conditions”, where is the discussion the causes and explanations of changes by increment or reduction and then we confirm the interpretations with previous studies.

According to your comments, the descriptions about the repeated sentences have been deleted or revised to: “Significant increase of plant biomass was observed at cooler temperature of 15°C than 22°C, which mainly resulted from the significant increase in root diameter (Fig. 2). The increase of plant biomass at 15°C authenticates that the A. sinensis species prefers to the cool environmental conditions, which is accordance with previous studies that higher-altitude improves the root yield and bioactive metabolites accumulation [11, 12, 14]” . (Page 7, lines 166-169)

In Materials and Methods:

Line 226, where is the centrifuge company and country made?

According to your comments, the centrifuge company and country made have been added: “TGL-20M, Changsha, China”. (Page 8, line 233)

Line 244, where is the spectrometer company and country made?

According to your comments, the spectrometer company and country made have been added: “(V1800, Shanghai, China) ”. (Page 9, lines 250 and 256)

Line 253, “ddH2O” should be cited as “dH2O”.

According to your comments, the “ddH2O”has been cited as “dH2O”. (Page 9, line 260)

In Conclusions:

Very weak conclusion, It must be one sentence to the most important results, sentence to results applications and whom will benefits from these results. A future outlook in the field of research, upcoming studies, and the beneficiaries thereof. Why reference in conclusion ?

According to your comments, the Conclusion has been revised to: “From the above observations, cooler temperature significantly enhances biomass accumulation, ferulic acid and flavonoid biosynthesis in A. sinensis as well as their related genes expression, while inhibits polysaccharide accumulation. These findings will provide a strong foundation for regulating plant growth and bioactive metabolites production of A. sinensis. The roles of sunshade and rainfall in the geo-authentic formation will be conducted in the upcoming studies”. (Page 10, lines 282-287)

Round 2

Reviewer 3 Report

I am satisfied now, the authors did my suggestions

Author Response

Thanks again for your suggestion and comments. we will carefully check the manuscript before publication.

Wish you happy new year!